# Construction of an Agricultural Drought Monitoring Model for Karst with Coupled Climate and Substratum Factors—A Case Study of Guizhou Province, China

Lihui Chen [1], Zhonghua He [1,2,3,*], Xiaolin Gu [4], Mingjin Xu [4], Shan Pan [1], Hongmei Tan [1] and Shuping Yang [1]

[1] School of Geography and Environmental Science, Guizhou Normal University, Guiyang 550001, China; chenlihui0828@163.com (L.C.)

[2] National Engineering Technology Research Center for Karst Rocky Desertification Control, Guizhou Normal University, Guiyang 550001, China

[3] Guizhou Key Laboratory of Remote Sensing Application of Mountain Resources and Environment, Guiyang 550001, China

[4] Guizhou Hydrology and Water Resources Bureau, Guiyang 550002, China

[*] Correspondence: zhonghuahe@edu.gznu.cn; Tel.: +86-15185008076

**Abstract:** Droughts are becoming more frequent in the karst region of southwest China due to climate change, and accurate monitoring of karst agricultural droughts is crucial. To this end, in this study, based on random forest (RF) and support vector regression (SVR) algorithms, the monthly precipitation, monthly potential evapotranspiration, monthly normalised difference vegetation Index (NDVI), elevation, and karst development intensity from January to December 2001–2020 were used as independent variables, and the standardised soil moisture index (SSI) calculated by GLDAS soil moisture was used as the dependent variable to construct karst agricultural drought monitoring models at different timescales, using Guizhou Province as an example. The performance of the models constructed by the two algorithms was also evaluated using root mean square error (RMSE), coefficient of determination ($R^2$), and correlation analysis, and the spatial and temporal evolution trends of karst agricultural drought at different timescales were analysed based on the model with better performance. The prediction of karst agricultural drought from January to December 2021–2025 was based on the seasonal difference autoregressive moving average (SARIMA) model and the analysis of change trends was performed using the Bayesian estimator of abrupt change, seasonal change, and trend (RBEAST). The results showed that (1) the drought model constructed by the RF regression algorithm performed better than the SVR algorithm at 1-, 3-, 6-, 9-, and 12-month timescales and was superior for monitoring karst agricultural drought. (2) The model showed that the overall trend of agricultural drought at different timescales was alleviated; 2010, 2011, and 2012 were typical drought years. At the same time, most regions showed a trend of drought mitigation, whereas a few regions (Bijie City, Liupanshui City, and Qianxinan Prefecture) showed a trend of aggravation. (3) The study predicted an overall high west–east distribution of drought intensity by 2021–2025. The 1- and 3-month timescales showed a trend of agricultural drought mitigation, and the 6-, 9-, and 12-month timescales showed a trend of aggravation; in 2021, 2022, and 2024, the abrupt change rates of autumn and winter droughts were higher. The results can provide a reference basis for the monitoring of agricultural drought in karst agriculture and the formulation of drought prevention and anti-drought measures.

**Keywords:** karst agricultural drought; monitoring models; random forests; support vector regression; mutation probabilities

## 1. Introduction

Drought is one of the most devastating natural disasters, affecting hydrology, meteorology, ecology, and society, and causing significant damage [1]. Droughts can be divided

into four categories [2]: meteorological drought, which occurs due to insufficient precipitation; agricultural drought, which occurs due to insufficient soil moisture and affects vegetation growth; hydrological drought, which occurs due to water shortage in rivers; and socio-economic drought, which occurs due to its impact on human life and regional development [3]. The complex interactions between vegetation and climate [4] make agricultural droughts more difficult to understand than other droughts, and because agriculture is a matter of national food security and is vulnerable to climate constraints [5], it is crucial to study agricultural droughts.

Many scholars have proposed many drought indices to characterise agricultural drought conditions based on station and remote sensing observation data, such as the Palmer drought severity index (PDSI) [6], the vegetation conditional index (VCI) [7], the temperature condition index (TCI) [8], the soil moisture condition index (SMCI) [9], etc. However, the formation mechanism of agricultural drought is complex [10], and it is difficult to accurately reflect drought conditions using a single drought index. Therefore, many scholars have combined remote sensing data with climatic factors affecting vegetation and other relevant environmental variables to construct an integrated drought model [11]. Sun Li et al. proposed the integrated monitoring drought index by linearly weighting the temperature vegetation drought index (TVDI) and the anomaly percentage of precipitation index and applying this to make the new index more stable than the TVDI [12]. Rhee proposed a scaled drought condition index by linearly combining NDVI, land surface temperature, and TRMM data [13]. These studies primarily constructed models based on linear weighting, and the drought-causing factors considered were limited. In recent years, scholars have attempted to construct drought models by considering multiple drought-causing factors, using methods such as machine learning. Brown et al. [14] considered vegetation anomalies, precipitation anomalies, and ecological environment parameters to construct a vegetation drought response index (VegDRI) using categorical regression trees. Based on VegDRI, Wu Jiaojiao et al. [15] proposed the integrated surface drought index with PDSI as the dependent variable for 14 factors characterising vegetation growth status, environmental water supply status, and soil conditions, and showed that it had greater advantages in agricultural drought monitoring. The synthesised drought index proposed by Du Lingtong et al. [16] achieved good results in drought monitoring at both regional and local scales. Sheng Runping et al. [17] constructed a remote-sensing drought-monitoring model based on multi-source remote-sensing data using a random forest (RF) algorithm and achieved good results in practical monitoring. The construction of integrated models to characterise agricultural droughts has become a trend.

In recent years, droughts in the southwestern karst region of Guizhou have become more frequent in the context of climate change [18] and have had an extremely severe impact on the region [19]. Many researchers have constructed comprehensive drought models to monitor drought; however, no studies, to our knowledge, have constructed agricultural drought models for karst regions. Therefore, this study used Guizhou Province as an example. We considered the precipitation, evapotranspiration, NDVI, elevation, and karst development intensity [20] and used machine learning algorithms (RF and SVR) to construct karst agricultural drought monitoring models at different timescales. This study also compared the performance of the two models, and the model with better performance was selected as the karst agricultural drought monitoring model. Based on this, Sen trend analysis and MK mutation tests were applied to explore the characteristics of the spatial and temporal distribution of agricultural drought in karst areas at different timescales. The SARIMA model was used to predict the drought situation for the next 5 years, while the RBEAST model was used to analyse the change trend of agricultural drought in karst areas for the next 5 years to provide a scientific basis for agricultural drought monitoring and drought mitigation.

## 2. Study Area

Guizhou Province is located in southwestern China (Figure 1), on the eastern slope of the Yunnan-Guizhou Plateau (103°36′–109°35′ E, 24°37′–29°13′ N), with a province area of about $1.76 \times 10^5$ km$^2$ and karst areas accounting for about 73.8% of the province's area. The landform types are mainly plateaus, mountains, hills, and basins, and it is the most widely distributed karst landform in the world [21]. There is a high river density in the territory, with a total length of 11,270 km, and the Wumeng Mountain Miaoling watershed belongs to the Yangtze River and Pearl River basins. The climate type is a subtropical humid monsoon climate [22], with an average annual temperature of 14–16 °C in most areas and an annual precipitation of 1100–1400 mm. However, there is uneven spatial and temporal distribution, with 75% of precipitation concentrated between April and September and a spatial trend decreasing from east to northwest, which leads to a more common drought phenomenon. The average elevation is 1100 m, with the terrain being high in the west and low in the east, sloping from the centre to the north, east, and south, and the overall frequency of droughts shows a similar distribution pattern. Drought was significantly aggravated by the El Niño phenomenon in 2011; in particular, karst development was intense, surface water storage capacity was weak, and soil erosion was extensive, affecting agricultural production.

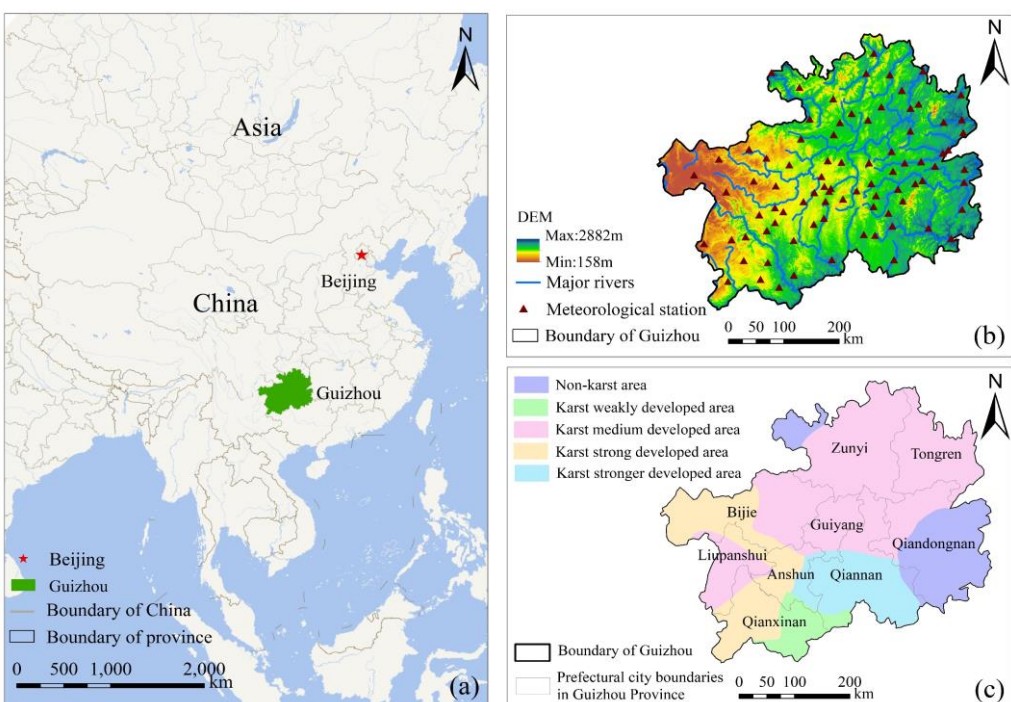

**Figure 1.** Location map of the study area in Guizhou Province, China. (**a**) The location of Guizhou Province in China; (**b**) elevation and the meteorological stations and main water systems in Guizhou Province; (**c**) karst development zoning in Guizhou Province.

## 3. Materials and Methods

### 3.1. Data Source and Pre-Processing

The data period of this study was January–December 2001–2020, and the projections were all unified as WGS-1984-UTM-zone-48N. The soil moisture data were obtained from the Global Land Data Assimilation System (https://ldas.gsfc.nasa.gov/gldas/#, accessed on 1 September 2021). The 0–10 cm data in kg/m$^3$ from 'GLDAS-NOAH025-M-2.1' were obtained from the NDVI of the LAADS DACC Data Centre (https://ladsweb.modaps.eosdis.nasa.gov, accessed on 1 September 2021). The precipitation and potential evapotranspiration data were obtained from the National Earth System Science Data Sharing Service Platform (http://www.geodata.cn, accessed on 1 September 2021). The elevations were obtained from the Data Centre for Resource and Environmental Sciences at the Chi-

nese Academy of Sciences (http://www.resdc.cn, accessed on 1 September 2021). The karst development intensity zoning was derived from the karst development intensity map of Guizhou Province in 'The Hydrogeology of Guizhou Province', which was then digitised and divided into non-karst, weakly developed, moderately developed, more strongly developed, and strongly developed zones. The NDVI, precipitation, and potential evapotranspiration were used to reflect their respective anomalies using a standardised method [23] (spatial resolution: 1 km, temporal resolution: 1 month) and were then extracted for subsequent studies based on 84 meteorological stations in Guizhou Province.

### 3.2. Research Methodology

In this study, using monthly precipitation anomalies (PreA), monthly potential evapotranspiration anomalies (PetA), monthly NDVI anomalies (NDVIA), and elevation (DEM) karst development intensity as independent variables and SSI as dependent variables, a karst agricultural drought model was constructed using RF and SVR. the model performance was assessed by RMSE, $R^2$, and standard deviation, and the correlation coefficients between the model and SPI were also analysed to select the model with better performance for monitoring karst agricultural drought. Based on this, the characteristics of karst agricultural drought were explored. The flow chart of this study is shown in Figure 2.

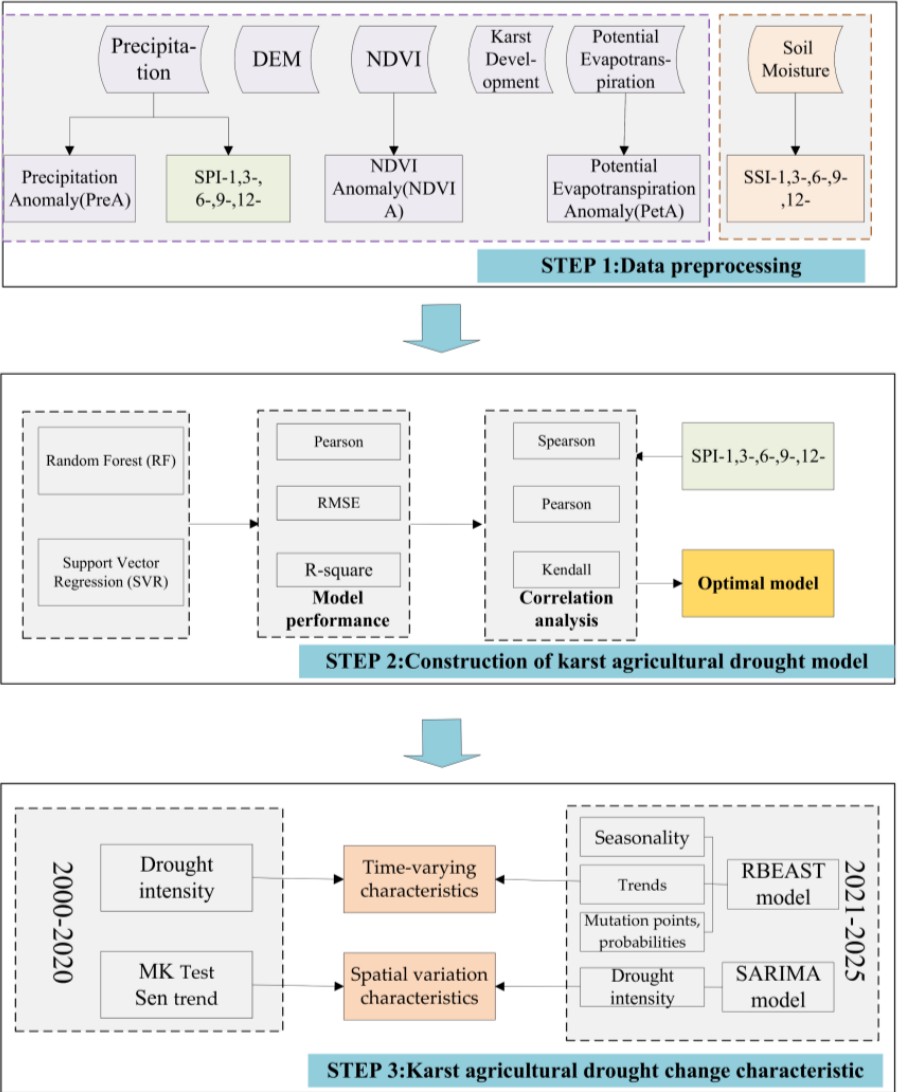

**Figure 2.** The flow chart of this study.

### 3.2.1. Drought Identification

Standardised indices are simple to calculate and can characterise the severity of drought at different timescales [24]; therefore, the standardised precipitation index (SPI) and standardised soil index (SSI) were chosen to identify drought in this study. Both were calculated similarly, and the main calculation procedure was as follows:

The monthly precipitation (P) or monthly soil moisture (S) series was fitted using the following probabilistic statistical distributions:

$$SPI|SSI = \Phi - 1(F(Dk_n)) \tag{1}$$

where $\Phi - 1(F(x))$ is the standard normal inverse transform and $Dk_n$ is P or S and denotes the accumulation of P or S on different timescales, and is calculated as:

$$Dk_n = \sum_{i=0}^{k-1}(P_{n-i}|S_{n-i}), n \geq k \tag{2}$$

where n is the number of months calculated, k is the cumulative timescale, and k = 1, 3, 6, 9, and 12 for a total of five timescales.

Gamma distribution is often used to fit the P or S series:

$$g(x) = \frac{1}{\beta\Gamma(\alpha)}x^{\alpha-1}e\frac{-1}{\beta} \tag{3}$$

where $\Gamma(\alpha)$ is the gamma function, x is cumulative P or S, and $\alpha$ and $\beta$ are the shape and scale parameters, respectively, of the gamma function estimated by the great likelihood method. The detailed calculation method has been described in previous studies, and the SPI and SSI were graded according to the national drought rating criteria for the degree of drought (Table 1).

**Table 1.** Classification of SSI and SPI drought levels.

| SPI\|SSI | Drought Level |
|---|---|
| $-0.5 \leq SPI|SSI$ | Normal |
| $-1.0 \leq SPI|SSI < -0.5$ | Light |
| $-1.5 \leq SPI|SSI < -1.0$ | Moderate |
| $-2.0 \leq SPI|SSI < -1.5$ | Severe |
| $SPI|SSI < -2.0$ | Extreme |

### 3.2.2. Karst Agricultural Drought Model Construction

(1) Model construction

Agricultural drought is a phenomenon in which prolonged abnormal precipitation results in a shortage of soil moisture to meet the normal growth requirements of vegetation. This is accompanied by a constant loss of water from vegetation transpiration, which, in turn, causes stress on vegetation growth and economic losses [25]. Therefore, the agricultural drought process not only involves factors such as atmospheric precipitation, vegetation growth state, and soil moisture, but also has a relationship with evapotranspiration, which affects precipitation patterns and the spatial distribution of human activities, in addition to the intensity of karst development affecting soil water retention and water storage capacity [20]. Drought-causing factors have complex coupling relationships [26], thus, in this study, monthly SSI (SSI-1, SSI-3, SSI-6, SSI-9, and SSI-12) at different timescales was used as the dependent variable; monthly PreA, monthly PetA, monthly NDVIA, DEM, and karst development were used as independent variables; and a semi-empirical and semi-mechanical agricultural drought monitoring model was constructed using RF and SVR regression algorithms to simultaneously evaluate the performance of both models.

(2)    RF model construction

The RF algorithm is an integrated machine learning algorithm based on decision trees that can effectively avoid the decision tree overfitting problem [27], including classification and regression algorithms [28]. It generates ntree new sample sets by randomly selecting samples from the dataset with put-back, and each sample set constructs a regression tree. The regression tree grows branches by randomly selecting mtry independent variables from the independent variables at each node. All the regression trees formed an RF, and the prediction was the mean of the results of each regression tree. This study constructed an RF karst agricultural drought monitoring model in the form of RF-CDI = (PreA, PetA, NDVIA, DEM, and Karst development) for different monthly timescales (January–December) based on the R platform random forest package. ntree and mtry are two important parameters for constructing an RF agricultural drought monitoring model. Two important parameters for the model, ntrees, were determined with a minimum mean square error (MSE) (upper limit of 2000) (Table 2), and mtry was fixed at five (i.e., all five independent variables were involved in the branch growth of each tree). The detailed RF algorithm has been described in previous studies [27].

**Table 2.** Constructing RF model ntree parameters.

| | January | February | March | April | May | June | July | August | September | October | November | December |
|---|---|---|---|---|---|---|---|---|---|---|---|---|
| **RF-CDI1** | 1505 | 1802 | 538 | 280 | 1700 | 2000 | 1887 | 378 | 165 | 578 | 345 | 819 |
| **RF-CDI3** | 473 | 294 | 679 | 1888 | 374 | 1993 | 1499 | 1701 | 180 | 1056 | 1271 | 806 |
| **RF-CDI6** | 533 | 1449 | 1983 | 995 | 1963 | 1967 | 673 | 944 | 1048 | 1916 | 1962 | 533 |
| **RF-CDI9** | 334 | 1990 | 1475 | 1410 | 1797 | 1411 | 1809 | 1980 | 324 | 1194 | 1988 | 245 |
| **RF-CDI12** | 308 | 1349 | 1656 | 669 | 1656 | 1092 | 822 | 822 | 837 | 1092 | 1994 | 241 |

(3)    SVR model construction

SVR is a supervised machine learning algorithm [29] for classification and regression that constructs a hyperplane or a set of hyperplanes in a high-dimensional space [30], which is advantageous for solving nonlinear problems and can overcome the shortcomings of neural networks in dealing with nonlinear problems. To minimise the hyperplane distance from all the samples to the constructed hyperplane, regression problems are based on different kernel functions that map low-dimensional samples to higher dimensions to make them linearly separable [31]. In this study, based on the SVR regression algorithm, a radial basis kernel function (radial) was used to construct the model, and the kernel coefficient gamma and penalty coefficient cost had a great impact on the accuracy of the model [32], thus, cross-validation was used to determine the optimal parameters that fitted the samples (Table 3). Finally, this study constructed an SVR karst agricultural drought monitoring model in the form of SVR-CDI = (PreA, PetA, NDVIA, DEM, and Karst development) for different monthly timescales (January–December) based on the R platform e1071 package. The detailed algorithm has been described in previous studies.

**Table 3.** Constructing SVR model ntree parameters.

| | | January | February | March | April | May | June | July | August | September | October | November | December |
|---|---|---|---|---|---|---|---|---|---|---|---|---|---|
| **SVR-CDI1** | gamma | 2 | 1 | 1 | 2 | 1 | 1 | 0.1 | 0.1 | 1 | 1 | 1 | 1 |
| | cost | 3 | 2 | 2 | 2 | 1 | 2 | 1 | 4 | 2 | 1 | 1 | 1 |
| **SVR-CDI3** | gamma | 3 | 3 | 3 | 2 | 2 | 2 | 0.1 | 1 | 4 | 1 | 4 | 1 |
| | cost | 1 | 2 | 2 | 2 | 3 | 1 | 1 | 4 | 2 | 1 | 2 | 1 |
| **SVR-CDI6** | gamma | 3 | 2 | 2 | 1 | 3 | 2 | 0.1 | 0.1 | 4 | 1 | 3 | 4 |
| | cost | 2 | 2 | 2 | 1 | 2 | 1 | 1 | 4 | 2 | 2 | 2 | 4 |
| **SVR-CDI9** | gamma | 4 | 2 | 2 | 3 | 4 | 2 | 0.1 | 0.1 | 3 | 1 | 2 | 4 |
| | cost | 2 | 2 | 2 | 2 | 3 | 2 | 1 | 4 | 2 | 2 | 3 | 4 |
| **SVR-CDI12** | gamma | 3 | 2 | 3 | 3 | 4 | 2 | 1 | 0.1 | 1 | 1 | 0.1 | 4 |
| | cost | 2 | 3 | 2 | 2 | 2 | 3 | 0.1 | 3 | 2 | 2 | 4 | 4 |

(4)    Evaluation model method

The coefficient of determination ($R^2$), root MSE (RMSE), and correlation coefficient R (Pearson, Kendall, and Spearman) were used to evaluate the performances of the two models, from which the most suitable model for agricultural drought monitoring in karst areas was determined. The higher the $R^2$ and R and the smaller the RMSE, the better the performance of the model, indicating that it was more suitable for karst agricultural drought monitoring.

### 3.2.3. Drought Prediction

The integrated autoregressive moving average (ARIMA) model [33] is a commonly used time-series forecasting method with high forecasting accuracy for the short-term forecasting of nonstationary time series [34], usually denoted as ARIMA (p,d,q), with specific model references provided in previous studies [35]. The seasonal difference autoregressive moving average (SARIMA) model, usually denoted as SARIMA (p,d,q) (P,D,Q), is used for time-series problems with seasonality, where (p,d,q) is the non-seasonal component (i.e., the ARIMA model) and (P,D,Q) is the seasonal component, reflecting the cyclical nature of the time series [36]. The model is expressed by Equation (4):

$$\varnothing_p(L)A_p(L^s)\Delta^d\Delta_s^D y_t = \Theta_q(L)B_Q(L^s)u_t \tag{4}$$

where $y_t$ is the drought intensity at time point t of the time series simulated by the agricultural drought model at each timescale; p, d, and q are the non-seasonal autoregressive order, number of differences in the transformation of the non-seasonal time series into a smooth time series, and non-seasonal moving average order, respectively; P, D, and Q are the seasonal autoregressive, differential, and moving average orders, respectively; s is the length of the seasonal cycle; $\varnothing$ and A are the parameters of the p-order autoregressive term and the seasonal autoregressive term, respectively; $\Delta^d$ and $\Delta_s^D$ are the differential operator and seasonal differential operator, respectively; $\Theta$ and B are the q order moving average term parameters and the Q order seasonal cycle moving average term parameters, respectively; L is the lag operator; and $u_t$ is the noise component of the stochastic model [37].

This study predicted the drought intensity for January–December 2021–2025 at different timescales based on sites using the SARIMA model, and the construction process was as follows: first, the augmented Dickey–Fuller test was used to identify the smoothness of each time series, and the trend difference and seasonal difference were used to convert the non-smooth data into smooth data. Then, the optimal parameter sets and model structures for p, q, P, and Q were selected based on the autocorrelation function (ACF) and partial ACF of the above smooth time series, combined with the Akaike information criterion [38].

### 3.2.4. Other Methods

(1) The Theil–Sen median (Sen) trend analysis [39], a nonparametric trend degree method, was used to calculate the trend of change in karst agricultural drought for different timescales from 2001 to 2020, and the significance of the trend was tested using the Mann–Kendall statistical test. The advantage of Sen trend analysis is that the sample does not need to obey a certain distribution [40], which can reduce the interference of outliers using the following equation:

$$\beta = \text{Median}\left(\frac{x_j - x_i}{j - i}\right), \, j > i \tag{5}$$

where $\beta$ is the drought trend and $x_i$ and $x_j$ are the drought intensities corresponding to i and j at a specific time in the study area. If $\beta > 0$, the agricultural drought tended to be alleviated, and if $\beta < 0$, it tended to be aggravated. A 0.01 significance level determined by the Mann–Kendall test indicated a highly significant change in drought, 0.05 indicated a significant change in drought, and 0.1 indicated a slightly significant change in drought.

(2) RBEAST is a traditional Bayesian algorithm that uses prior information to infer the model structure [41] and it can be used to detect seasonality, trends, mutation points, and mutation probabilities in a time series. It does not rely on a single model to decompose the time series, but is based on an integrated algorithm that combines many weak models into a stronger model while treating all the noise as random [42]. This study analysed the trend of karst agricultural drought at different timescales from January–December 2021–2025 based on the RBEAST model using the following model expressions:

$$Y(t) = T(\theta_t) + S(\theta_s) + \varepsilon \tag{6}$$

where Y(t) is the time series; T and S are the trend and seasonal terms, respectively; $\theta_t$ and $\theta_s$ are the number of change points in the trend term and the location of change points in the seasonal term, respectively; and $\varepsilon$ is a Gaussian random error term N $(0, \delta^2)$ with unknown variance $(\delta^2)$. The specific calculation method has been described in previous studies [43].

## 4. Results and Analysis

### 4.1. Performance Evaluation of the Karst Agricultural Drought Monitoring Model

4.1.1. Model Validation and Evaluation

A total of 60 drought models were constructed for 1-, 3-, 6-, 9-, and 12-month timescales using RF and SVR algorithms to model the January–December Pearson correlation coefficients, RMSE, ratio of standard deviations, and $R^2$ (Figures 3 and 4), which were calculated for the dependent variable (SSI) and model fit values (RF-CDI and SVR-CDI) to compare the performance of the two types of agricultural drought models. In terms of the correlation coefficients, those with a total of 60 RF-CDI simulations with SSI values for different timescales from January to December ranged from 0.750 to 0.990, and were mainly concentrated above 0.9, with 50 models having correlation coefficients of >0.950 with SSI values, while the correlation coefficients of the all SVR-CDI fit with SSI values ranged from 0.380 to 0.950 with a wide range of fluctuations. The correlations of the January–December models for each timescale of the two algorithms passed the 0.01 significance test, and the simulation results of RF-CDI were superior. On a 1-month timescale, for example, the correlation coefficients of the RF-CDI 12-month model were all higher than those of the SVR-CDI model. In addition, the correlation of the RF-CDI was similar to that of the 1-month timescale for each month model for all timescales, except for the 9- (0.758) and 12-month timescales (0.788), where the correlation of the 5-month model was lower than that of the SVR-CDI (9:0.954, 12:0.938), and the correlation coefficient of the RF-CDI was higher than that of the SVR-CDI. In terms of the ratio of standard deviation, RF-CDI was concentrated in the range of 0.3 to 0.4, while SVR-CDI was concentrated in the range of 0.4 to 0.5 but fluctuated more. The RF-CDI simulation performed better. In terms of the RMSE between the simulation results and the SSI values, the RF-CDI remained at 0.190–0.290 for each timescale, except for July, when the RMSE was higher (>0.3), and other months when the RMSE was higher (>0.7). Again, the RF-CDI simulation was superior. The $R^2$ value of the July model was lower than that of the other months at the same scale for both RF-CDI and SVR-CDI (Figure 4). The $R^2$ value was like to that of the RMSE pattern, and the RF-CDI $R^2$ value was always maintained at a high level ($R^2 > 0.900$), except for July, whereas that of SVR-CDI fluctuated more, and both were lower than that of the RF-CDI. In conclusion, the performance of the karst agricultural drought model constructed based on the RF algorithm was better, and it could better monitor karst agricultural drought.

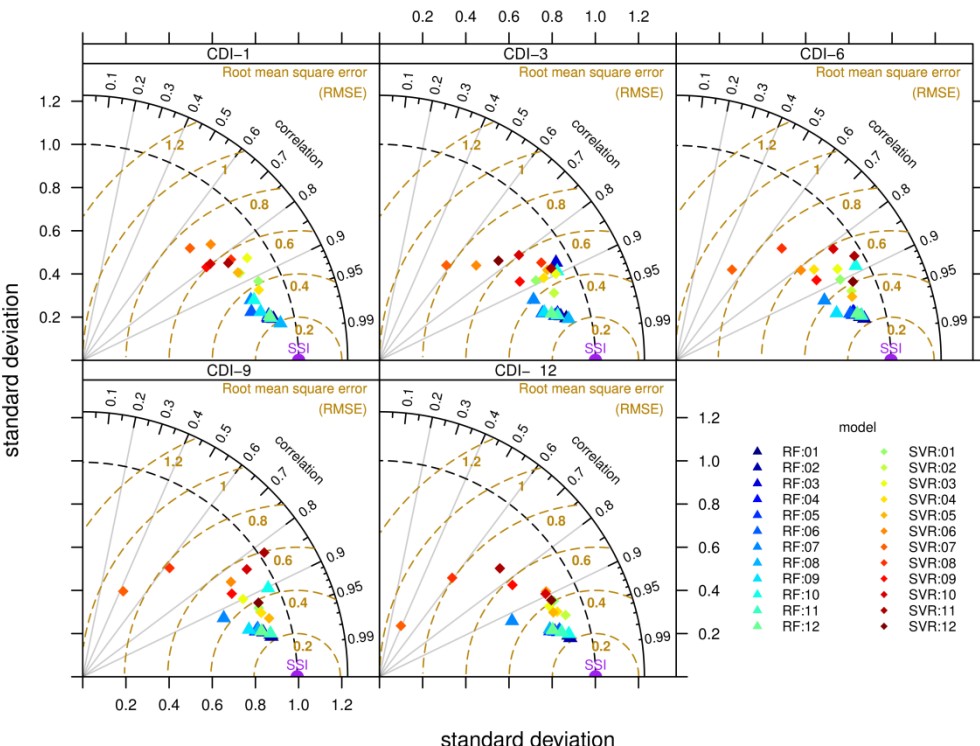

**Figure 3.** Taylor plots of model performance at different time scales constructed by the two algorithms. Note: Scatter is the model, the horizontal coordinate is the standard deviation of station SSI, the vertical coordinate is the standard deviation of model predictions, the radial line is the Pearson correlation coefficient, and the dashed line is RMSE.RF:01 represents the January drought monitoring model constructed by RF; other models are similar.

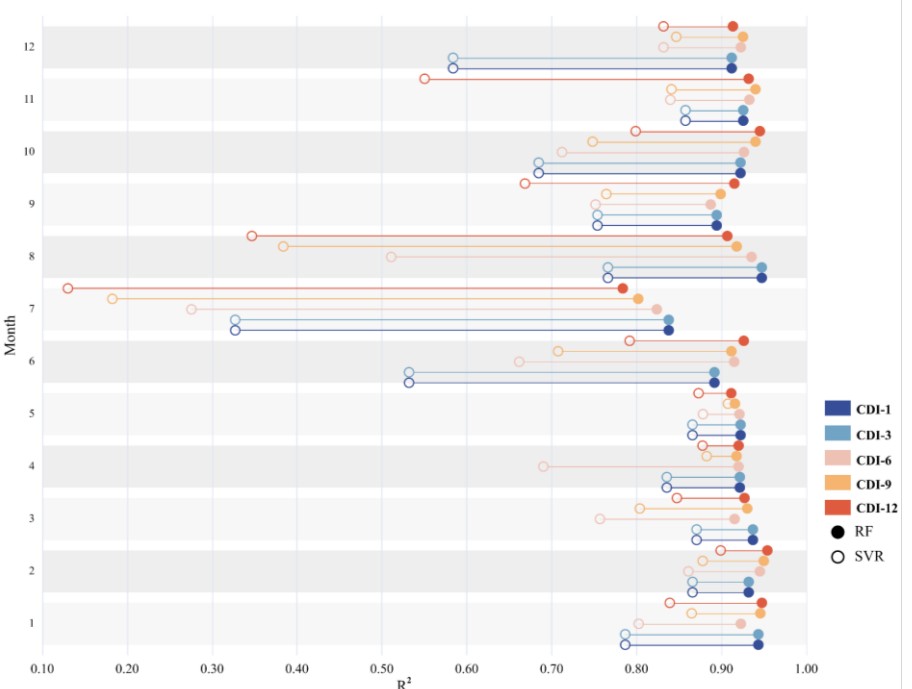

**Figure 4.** $R^2$ diagrams of different time scale models constructed by RF and SVR. Note: Solid circles indicate the model constructed by RF, while open circles indicate the model constructed by SVR.

### 4.1.2. Correlation Metric of RF-CDI, SVR-CDI, and SPI

In this study, the SPI was used to further evaluate the performance of RF-CDI (model constructed by RF) and SVR-CDI (model constructed by support vector institutions). Correlation analysis (Pearson, Kendall, and Spearman) was performed between RF-CDI and SVR-CDI and SPI at different timescales, and the results are shown in Figure 5 for the Pearson correlation analysis (pRF-CDI and pSVR-CDI). The correlation coefficient of SIP-9 with RF-CDI on a 1-month timescale (pRF-CDI-1), for example, was 0.547, while that of the corresponding SVR-CDI was 0.435. In addition, the correlation coefficient of pRF-CDI-12 with SPI-1 was smaller (0.162), and that with pSVR-CDI-12 was even smaller (0.159). The Pearson correlation coefficients of RF-CDI with SPI at different timescales were both higher than those of SVR-CDI with SPI at the corresponding timescales, and both passed the significance test at $p < 0.01$. For the Kendall correlation analysis, the pattern of correlation coefficients was similar to that of the Pearson correlation analysis, with all timescales passing the significance test of $p < 0.01$, except for the correlation coefficients between RF-CDI and SPI-1 at the 6-, 9-, and 12-month timescales, which passed the significance test of $p < 0.1$. The patterns of the Spearman correlation analysis and Pearson correlation analysis were similar (Figure 5). This further indicated that RF-CDI is superior for monitoring agricultural droughts in karst areas.

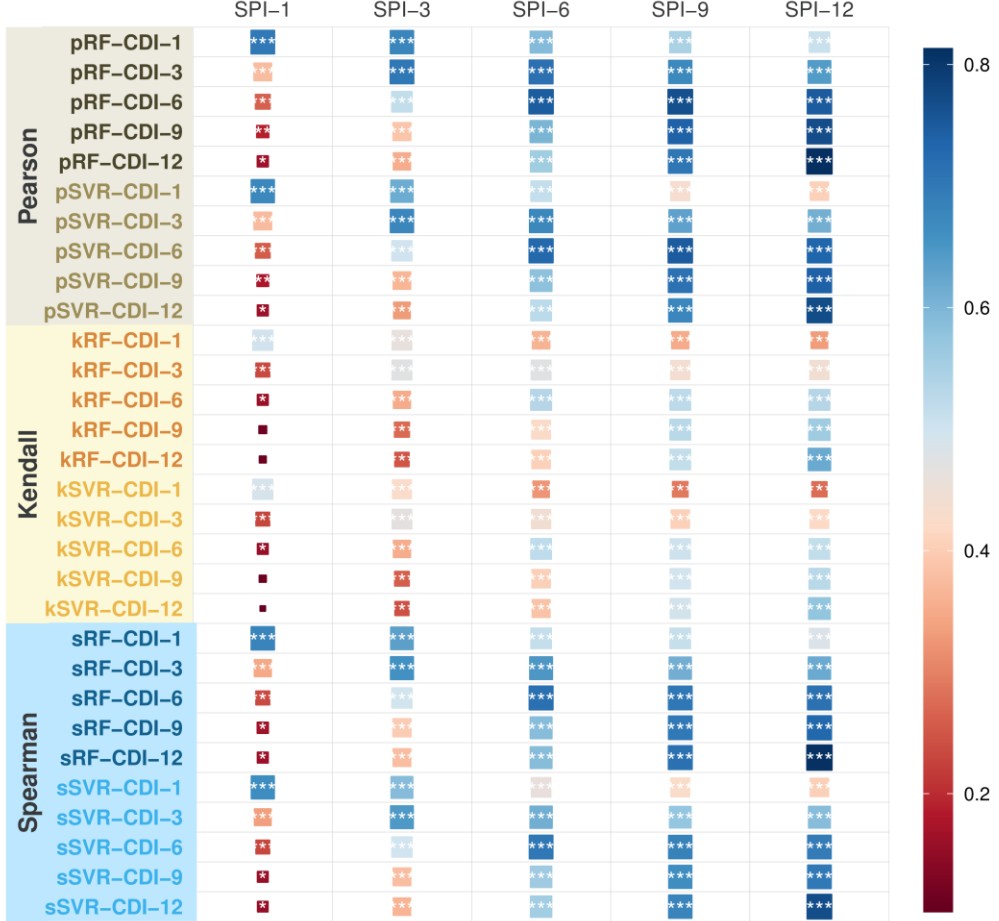

**Figure 5.** Correlation between model and SPI. Note: *p* in pRF-CDI-1 is the Pearson correlation coefficient, RF is the random forest, and 1 is the time scale; refer to this expression for other correlation coefficients; *** represents passing a significance test of 0.01, ** passing a significance test of 0.05, * passing a significance test of 0.1.

*4.2. Analysis of Karst Agricultural Drought Change Characteristics*

4.2.1. Time-Varying Characteristics

This study characterised karst agricultural drought based on RF-CDI for different timescales (1, 3, 6, 9, and 12 months) in Guizhou Province from 2001 to 2020 and analysed the temporal variation characteristics (Figure 6). Figure 6 shows that droughts on different timescales have different oscillation frequencies and exhibit different dynamic characteristics. For the 1-month timescale (RF-CDI-1), the drought characteristic values showed a fluctuating upward trend (tilt rate: 0.014/10 m), indicating an overall weakening trend of the drought intensity, with an average of −0.044. Taking 2011 as a typical year, the drought level in April, July, August, and September was extremely high, with severe drought in May (−1.737). March 2010 showed an exceptional drought level, and February of the same year showed severe drought. RF-CDI-3 (0.020/10 m) fluctuated in a similar trend to RF-CDI-1, and the intensity of drought tended to weaken. Similarly, 2011 and 2010 were typical drought years, with an exceptional drought in the summer and autumn of 2011, and severe drought in May, June, and November. February, March, and April 2010 experienced more severe drought. RF-CDI-6 also showed a fluctuating upward trend (0.023/10 m), and unlike the 3-month timescale, the drought that occurred in 2011 was more severe, with five of the six months experiencing the extreme drought occurred in 2011, affecting the summer, autumn, and winter (August, September, October, November, and December).

The same was true for the spring (March, April, and May) drought in 2010, during which the drought intensity in March reached −2.025, indicating that it was an extreme drought. RF-CDI-9 showed a greater fluctuation trend than the previous three timescales, but the overall trend was also upward (0.025/10 m), including during the severe drought in 2010 and 2011. The drought in February, March, and April 2012 was also severe, with the different months showing extreme drought, severe drought, and severe drought, respectively. RF-CDI-12 had a fluctuation trend similar to that of RF-CDI-9 (0.025/10 m). Drought was severe in the winter of 2011 and in the first half of 2012.

4.2.2. Spatial Variation Characteristics

To further explore the characteristics of karst agricultural drought change, this study analysed the change trend of monthly RF-CDI at different timescales from 2001 to 2020 using the Theil–Sen Median method and tested the significance of the change trend using the Mann–Kendall statistical test (Figure 7). The area with a positive slope of change of RF-CDI-1 accounted for approximately 95.506% of the total area (Figure 7a), indicating that the change trend of agricultural drought in Guizhou Province was mainly alleviated with a maximum value of 0.031/10 m, but 30.952% of the stations did not show a significant increase (did not pass the significance test) and were mainly distributed in the southwest region of the study area (Bijie City, Liupanshui City, and Qianxinan Prefecture). In addition, 25% of the stations had a highly significant upward trend, mainly in the Middle Eastern region (Meitan, Guiding, Majiang, Rongjiang, etc.), and 23.910% and 19.048% of the stations had significant and slightly significant upward trends, respectively. Among the 84 stations in the province, only one station in Panxian showed a decreasing trend, but did not pass the significance test. The spatial distributions of the RF-CDI-3 and RF-CDI-1 change trends were similar (Figure 7b), with the province's agricultural drought change trends dominated by mitigation (94.382%), and with sites that did not pass significance concentrated in the southwest region, but with a maximum value of up to 0.040/10 m. Compared to the previous two timescales, the number of significant sites in RF-CDI-6 (83.333%) has increased (Figure 7). The trend of RF-CDI-9 was similar to that of RF-CDI-6, with the exception of the northeastern station of Songtao, which had a non-significant increase. The number of stations that did not pass the significance test (10) was significantly reduced compared with other timescales, and most of them were highly significant (57). In conclusion, the overall trend of karst agricultural drought at different timescales from 2001 to 2020 was alleviated.

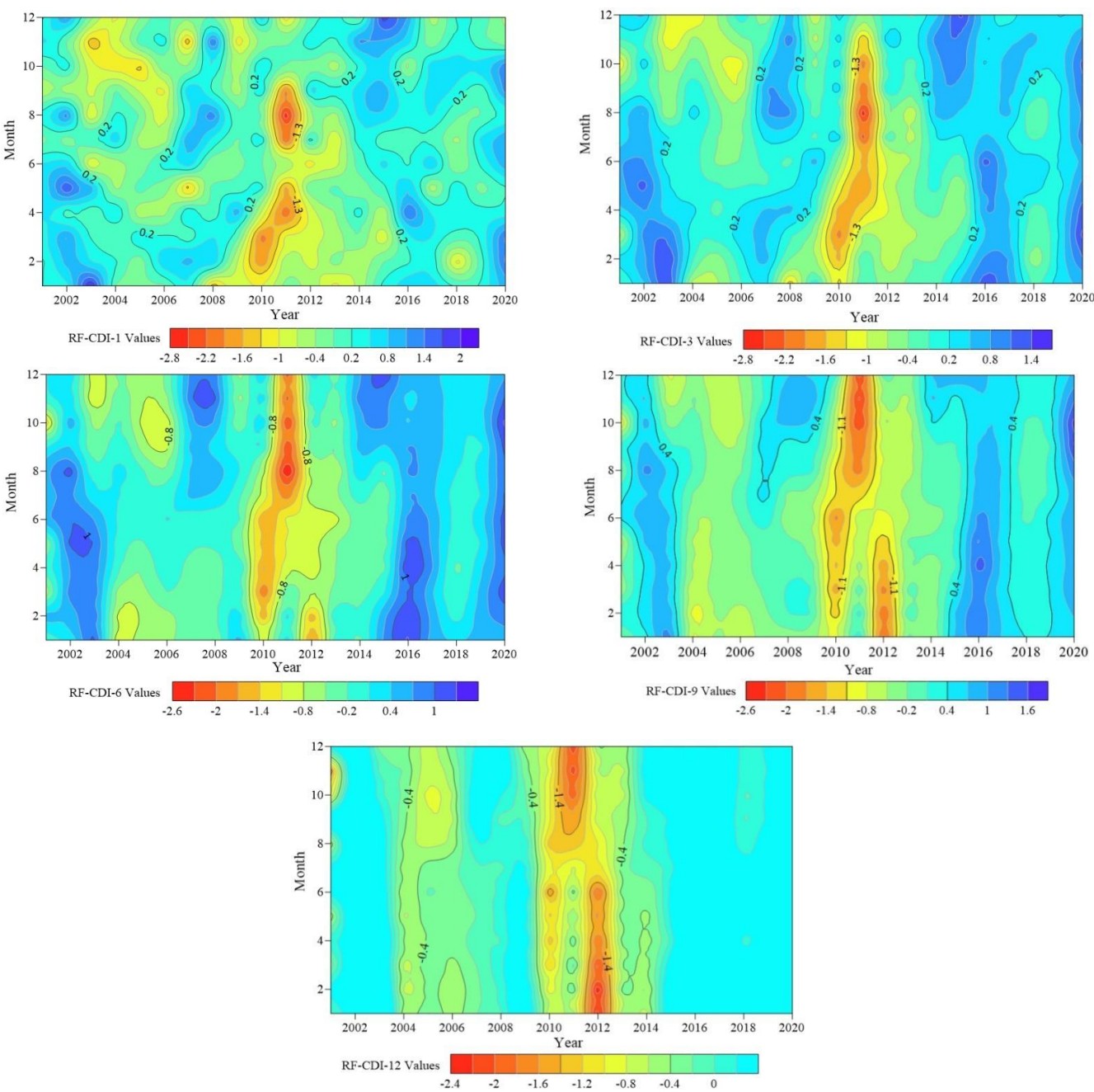

**Figure 6.** RF-CDI values at different time scales, January–December 2001–2020.

### 4.3. Karst Agricultural Drought Forecast for the Next 5 Years

4.3.1. Spatial Distribution Characteristics of Karst Agricultural Drought in the Next 5 Years

In this study, the SARIMA model was used to predict agricultural droughts at different timescales from January to December 2021–2025, and since August is prone to drought, the spatial distribution of agricultural droughts in karst from August 2021–2025 was plotted (Figure 8) and analysed. For the 1-month timescale, the spatial distribution pattern of August drought in Guizhou Province over the next 5 years is similar, and the overall drought characteristic values are low in the southwest and high in the east, implying that the drought in the southwest region of the study area is more severe than in other regions during the same period. However, the most severely affected region is northeast of Tongren (Yanhe, Yinjiang, and Songtao), and the most severe drought occurs in August 2021 as a light drought (−0.563). The 3-month timescale has a similar spatial distribution pattern as

the 1-month timescale: the drought intensity is higher in the west and lower in the east, but there is a local drought centre. The lowest drought characteristic value in August 2021 was −0.748 (Weining), and the drought intensity was more severe than the 1-month timescale in the same period; however, there were no drought areas in the following 4 years. Compared with the first two timescales, the 6-month timescale showed an overall west-high east-low distribution of drought intensity, but the local drought was more severe. The 9-month timescale was the most severe timescale of local drought, except for the severe drought in Weining, and the intensity of drought in Songtao reached −1.569 (severe drought) in August 2021. The following 4 years were similar to the 6-month timescale, with moderate drought. The 12-month timescale was similar to the 9-month timescale; however, the local drought (Guiyang) was relieved, and no drought occurred in the study area in August 2021. In summary, the spatial distribution of agricultural drought at different timescales over the next 5 years was variable, but the drought intensity as a whole showed a high distribution in the west and a low distribution in the east.

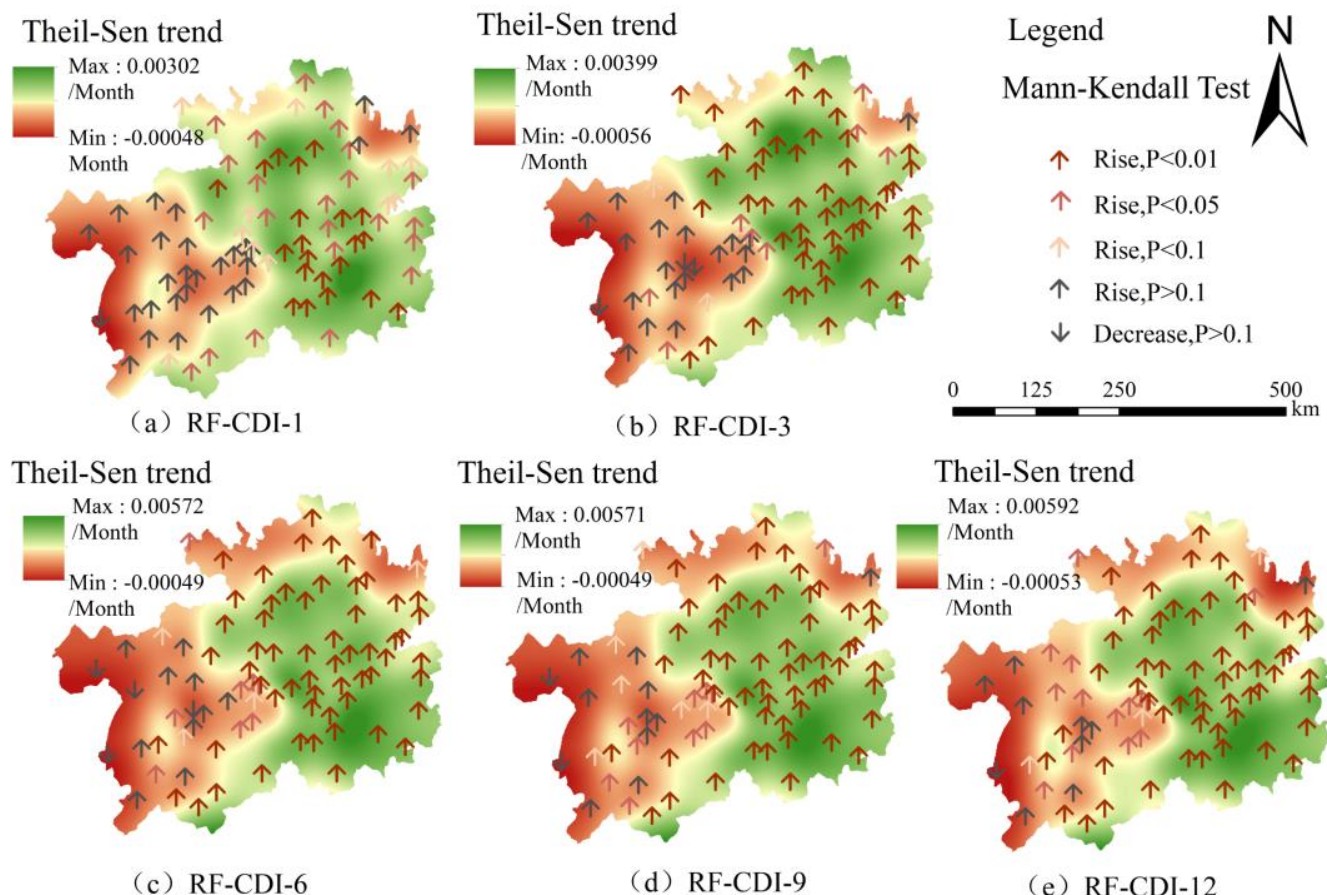

**Figure 7.** Spatial distribution of Sen trend + MK test at different time scales, January–December 2001–2020.

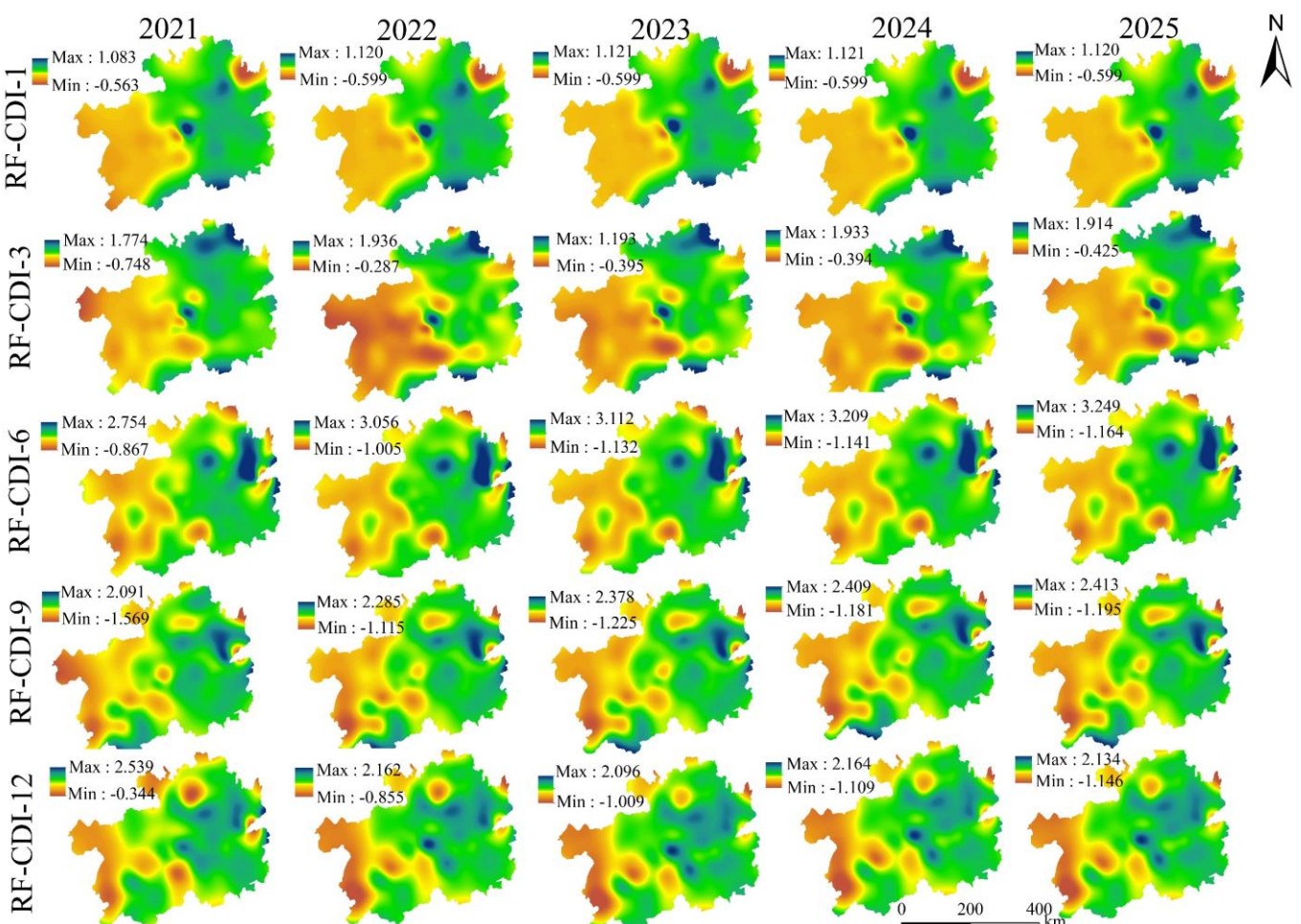

**Figure 8.** Spatial distribution of agricultural drought at different time scales, August 2021–2025.

### 4.3.2. Characteristics of Temporal Changes of Karst Agricultural Drought for the Next 5 Years

To further describe the future trends of agricultural drought in karst, this study used RBEAST to analyse the seasonality, trends, mutation points, and mutation point probabilities of agricultural drought at different timescales from January to December 2021–2025. The results show that agricultural drought for the next five years will have a trend of mitigation at the 1- and 3-month timescales and aggravation at the 6-, 9-, and 12-month timescales (Figure 9). For the 1-month timescale, there were two significant abrupt change points (October 2021 and January 2023) in the seasonality of drought characteristic values (Figure 9a) with a probability of abrupt changes of 100% and 88.596%, respectively, and an abrupt change, but with a lower probability in October 2022. With regard to the trends, there was generally an increase followed by a slight decrease, and then a levelling off. First, the agricultural drought was significantly relieved from January to November 2021, followed by a slight increase until October 2022, after which it levelled off. This meant that there were also two mutation points in the trend, which were November 2021 (probability, 92.201%) and October 2022 (73.858%); also in addition, Figure 9a shows that there is a higher probability that the agricultural drought was relieved in January–November 2021, followed by a higher probability that the agricultural drought was aggravated until October 2022. Following that, the agricultural drought seasonality was more variable on a 3-month timescale than on a 1-month timescale, but the overall agricultural drought tended to ease. Seasonality exhibited three abrupt change points: November 2021 (100%), August 2022 (86. 675%), and February 2024 (70.142%). There were two mutation points with a high probability of trends (November 2021, 89.013%; January 2024, 63.571%), with a trend of drought remission followed by slight aggravation. Two mutation points existed for 6-month timescale seasonality in July 2021 (100%) and September 2022 (76.358%).

The trend was different from that of the previous two timescales, with the drought showing an increasing trend from January to July 2021, a sudden change in July (96.063%), relief in November 2022, another sudden change in November 2022 (96.061%), and an increase in drought until March 2024 (79.481%), when the drought stabilised after a relief trend. The 9-month timescale was similar to the 6-month timescale, with an overall increasing trend of drought. Regarding the seasonality, there were three abrupt change points: July 2021, December 2022, and December 2023. Drought showed an increasing trend from January to August 2021, followed by a decreasing trend from August 2021 to October 2022, and then stabilised after an increasing trend from October 2022 to January 2024. The 12-month timescale seasonality showed abrupt changes in January 2022, July 2022, and August 2023. Overall, drought showed aggravation-slight, mitigation-slight, and aggravation-stabilisation trend, with abrupt changes in September 2021, April 2022, and July 2023.

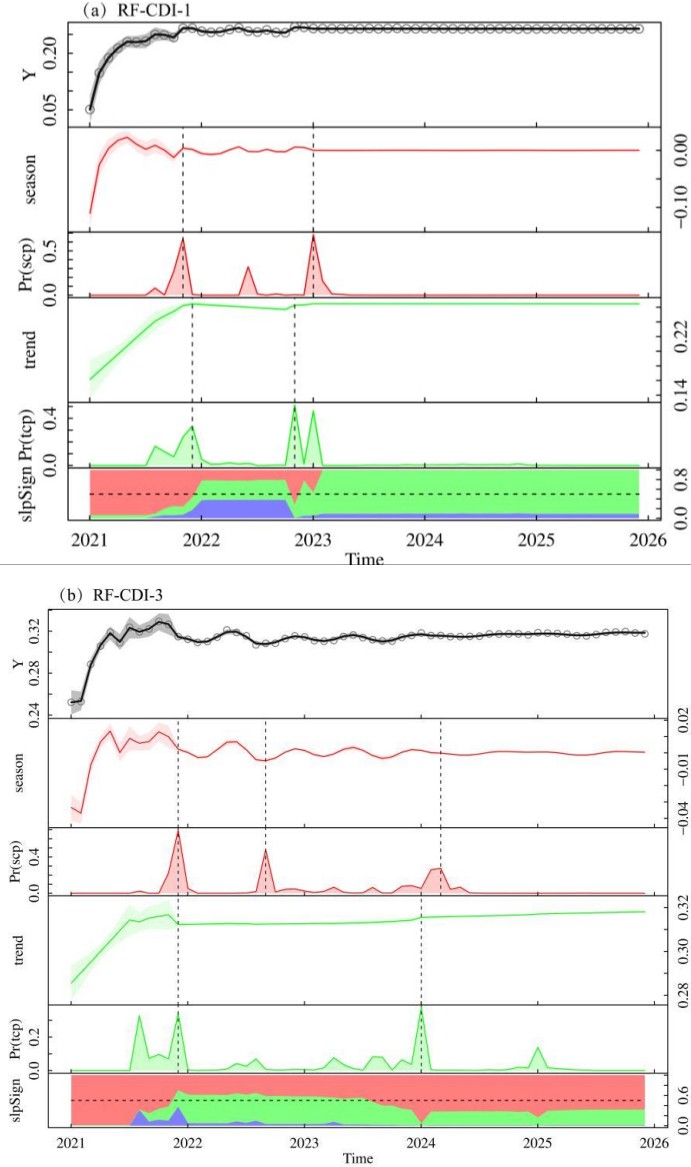

**Figure 9.** *Cont.*

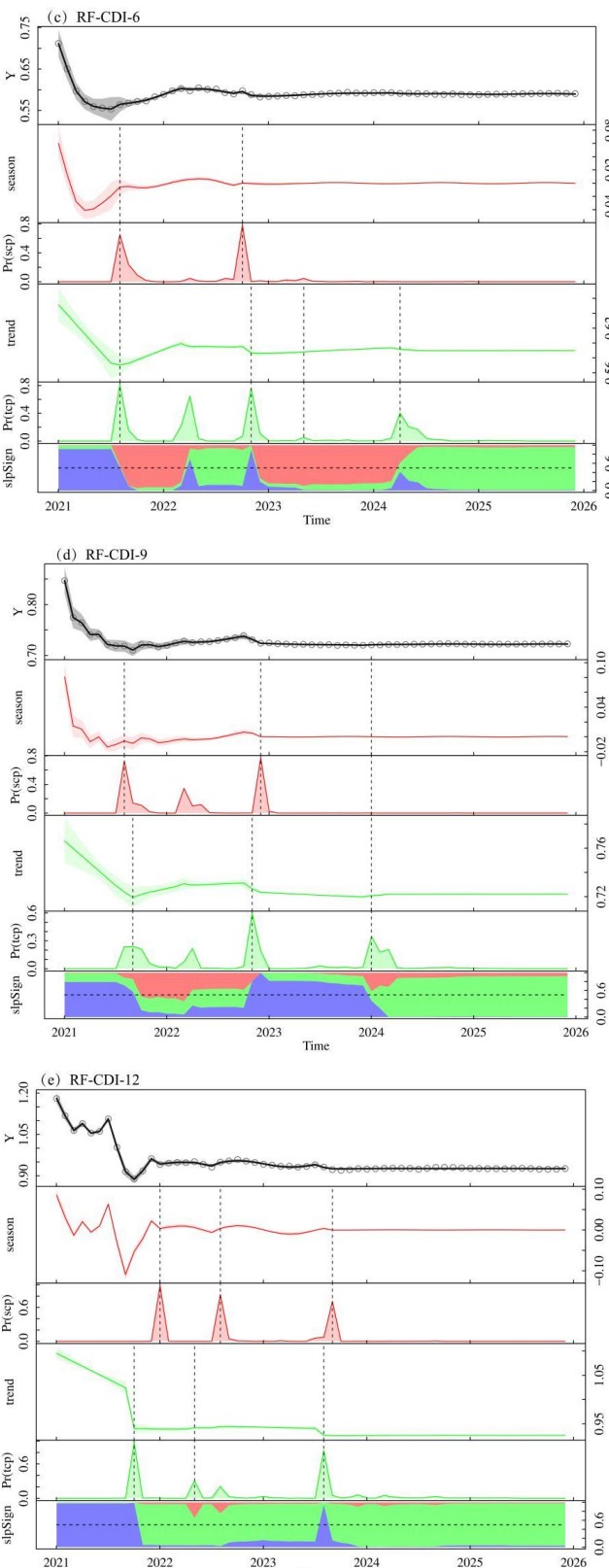

**Figure 9.** RBEAST mutation tests for agricultural drought at different time scales. Note: figure Y indicates agricultural drought intensity; season indicates seasonality; Pr(scp) indicates seasonal abrupt change points and probability; trend indicates a trend, pr(tcp) indicates trend abrupt change points and probability; and slpSign indicates a trend of change. Purple indicates drought relief, red indicates aggravation, green indicates no change.

## 5. Discussion

Precipitation is a key process in the global water cycle [44], and drought is inseparable from the water cycle process [45]. Guizhou Province has a subtropical humid monsoon climate that is alternately controlled by tropical marine air masses and polar continental air masses with an abundant but regional distribution of precipitation. In spring (March, April, and May), surface vegetation that is sprouting requires more water; vegetation sprouting is affected, and vegetation growth is inhibited in areas with precipitation deficits. In summer (June, July, and August), high temperatures increase vegetation transpiration. When precipitation cannot be effectively replenished, the vegetation close their leaf stomata or even shed old leaves to prevent excessive water evaporation [46]. In autumn (September, October, and November), surface crops mature and require less water than in spring, but reduced rainfall and persistent high temperatures can affect water vapour transport, and thus, vegetation growth. However, winter (December, January, and February) has low precipitation and high relative wetness and snowfall, resulting in less river recharge than in other seasons, which can trigger an agricultural drought.

Secondly, karst development and elevation constitute different subsurface conditions, which affect the water storage capacity of the basin. The unique 'surface-subsurface binary structure' of karst landscapes has a stronger water storage capacity in the basin compared with that of non-karst areas. The higher the elevation, the farther the surface is vertically from the basin's dissolution/erosion datum, and the greater the water storage space and water storage capacity [47]. When vegetation growth is inhibited by insufficient precipitation, the watershed water storage capacity is weak and vegetation cannot be effectively recharged, thus accelerating the occurrence of drought. At the same time, high altitudes can lead to low temperatures, which is also a major factor limiting the growth of vegetation. Therefore, the agricultural drought process is extremely complex and not only involves factors such as atmospheric precipitation, evapotranspiration, soil moisture, and the vegetation growth state, but is also related to the substratum (elevation and karst development intensity), which is the result of multi-factor coupling. In summary, this study clarified the coupling of drought-causing factors and constructed an agricultural drought model based on RF and SVR regression algorithms for karst areas, and the model constructed by the RF algorithm showed a better performance and was thus finally selected to monitor karst agricultural drought. The chosen RF algorithm was constructed based on a regression tree, which is an integrated learning method that can achieve information superiority and has a higher fitting accuracy than the SVR algorithm (Figures 3–5). It is more suitable for karst areas, providing a reference basis for karst agricultural drought monitoring.

In addition, this study found an overall trend of alleviation of agricultural drought in Guizhou Province at different timescales (1, 3, 6, 9, and 12 months) over the past 20 years (Figures 6 and 7), which may be attributed to the significant improvement in vegetation cover in Guizhou Province in recent years through the implementation of various measures, such as returning farmland to forest and rock desertification management [48], which is consistent with the findings of a study by Pi et al. [49]. Overall, agricultural drought was more severe in 2010, 2011, and 2012 (Figure 5), and this study identified severe drought for 1 month in February ($-1.867$), exceptional drought in March ($-2.082$), and severe drought in summer 2011 (drought intensity: July, $-2.284$; August, $-2.764$; September, $-2.075$). Drought conditions were severe in spring 2010; summer, autumn, and winter 2011; and spring and summer 2012. The historical disaster record recorded a very severe summer, autumn, winter, and spring drought in 2010; an exceptionally severe summer, autumn, winter, and spring drought in 2011; and a drought in most of Qianxinan in 2012, which coincided with the monitoring results of this study. It is worth noting that from 2009 to 2013, against a background of atmospheric circulation anomalies, southern branch troughs, and persistently weak stratospheric polar vortices, the Arctic Oscillation anomaly changed the cold air path eastward, and the warm and humid airflow had difficulty reaching Guizhou Province, which is an important reason for the severe drought in the study area [50], further

verifying the results of this study. From the spatial variation characteristics (Figure 7), the drought in the southwest region of the study area showed an aggravating trend and was more severe than that in the east, which was related to the weak horizontal transport of westerly winds [50].

Finally, this study predicted that the overall drought intensity for the next 5 years will be high in the west and low in the east (Figure 8), with a similar spatial distribution to the altitude of Guizhou Province (Figure 1). This may be attributed to the high altitude, fragmented surface, and high permeability of karst landscapes in the west that are highly susceptible to high-intensity drought and serious stone desertification [45]. Farming in areas with large slopes becomes inevitable, making crops and soils more susceptible to drought [51]. In addition, the temperature decreases with increasing elevation, and low temperatures can limit vegetation growth. Meanwhile, it is predicted that agricultural drought will show a trend of alleviation at the 1- and 3-month timescales and aggravation at the 6-, 9-, and 12-month timescales for the next 5 years. This may be due to the long timescale, extended drought epoch, and gradual weakening of the water storage capacity of the watershed, which increases the probability of propagation between droughts and aggravates agricultural drought. In addition to the timing and probability of abrupt changes at each timescale (Figure 9), this study could provide a reference for the development of drought prevention and relief measures in Guizhou Province. In addition, this study predicts a high probability of sudden changes in karst agricultural drought by autumn and winter in 2021, 2022, and 2024. Consulting relevant information from the Guizhou Agricultural Meteorological Bureau, the average autumn temperature in 2021 was 2.4–6.7 °C higher than in the rest of the year, with low precipitation and rapid development of drought, frequent cold air activity in late autumn and early winter, and low precipitation before winter compared with the same period, with insufficient soil subsoil moisture. A severe drought was followed by a cold wave in the autumn and winter of 2022. The accuracy of this model for monitoring drought in karst agriculture can be verified to some extent. The study focused on clarifying and modelling the agricultural drought process under natural conditions; however, drought is a complex natural phenomenon [52], and human activities, national policies, and other influencing factors were not considered, which is a shortcoming of this study and will continue to be explored in the future.

## 6. Conclusions

In this study, five drought-causing factors (precipitation, evapotranspiration, NDVI, elevation, and karst development intensity) were considered, and a semi-empirical and semi-mechanical agricultural drought monitoring model for karsts was constructed using RF and SVR algorithms with SSI as the dependent variable. The performance of the model was evaluated using the Pearson correlation coefficient, RMSE, $R^2$, and correlation analysis with SPI (Pearson, Kendall, and Spearman). The drought model based on the RF algorithm was found to be superior for monitoring agricultural drought in karst areas.

Based on the RF-CDI, the spatial and temporal evolutionary characteristics of droughts in the region over a 20-year period were explored. In terms of the temporal changes, the model monitored the overall trend of agricultural droughts at different timescales to be in remission, with 2010, 2011, and 2012 being more severe and typical drought years. In terms of spatial variation, the regions with positive Theil–Sen trend values for agricultural drought at different timescales accounted for the majority of the study area, indicating that most regions showed a trend of drought mitigation, with a small number of regions showing a trend of aggravation, mainly in the southwest region of the study area (Bijie City, Liupanshui City, and Qianxinan Prefecture). Most stations passed the test at $p < 0.01$.

The spatial distribution of agricultural drought at different timescales over the next 5 years was variable, but the drought intensity as a whole had a high west–east distribution. At the same time, it was predicted that the agricultural drought for the next 5 years will follow the trend of mitigation at the 1- and 3-month timescales and the trend of aggravation

at the 6-, 9-, and 12-month timescales. The probability of abrupt changes in the autumn and winter of 2021, 2022, and 2024 is high.

**Author Contributions:** L.C.: Methodology, data curation, and writing—original draft. Z.H.: Conceptualization, methodology, project administration, and funding acquisition. X.G.: Editing and supervision. M.X.: Editing and supervision, S.P.: Editing and supervision. H.T.: Editing and supervision. S.Y.: Editing and supervision. All authors have read and agreed to the published version of the manuscript.

**Funding:** The authors are grateful to the editors and anonymous reviewers for their useful suggestions and comments. This study was supported by the Natural Science Foundation of China (ul612441; 41471032); the Natural Science Foundation of Guizhou Province, China (OKHJ-ZK 2023 Key028); the natural and scientific research fund of Guizhou Water Resources Department (KT202237); the Natural and scientific fund of Guizhou Science and Technology Agency (OKH J 20101 No. 2026, OKH (20131 No. 2208)); and the 2015 Doctor Scientific Research Startup Project of Guizhou Normal University.

**Data Availability Statement:** The data that support the findings of this study are available from the corresponding author upon reasonable request.

**Conflicts of Interest:** The authors declare no conflict of interest.

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
