# Peer review of "Construction of an Agricultural Drought Monitoring Model for Karst with Coupled Climate and Substratum Factors—A Case Study of Guizhou Province, China"

_water, doi:10.3390/w15091795_

Round 1
Reviewer 1 Report
Currently, climate change has led to increased drought in the karst region of southwest China, however, there are few studies on accurate monitoring of karst agricultural drought. To this end, the authors took Guizhou province as an example, and constructed and analyzed and evaluated karst agricultural drought monitoring models at different time scales based on RF and SVR algorithms using monthly precipitation and NDVI from January to December 2001-2020 as independent variables , and SSI calculated by GLDAS soil moisture as dependent variables. The study predicted that by 2021-2025, the drought intensity generally showed a high west-east distribution; the abrupt change rates of both autumn and winter droughts were relatively high in 2021, 2022 and 2024. This study constructs a model based on the previous 10 years of data, which is important in predicting future drought conditions in the region perhaps should be beyond 2023. However, we are currently in the year 2023, and the "typical drought years of 2010, 2011 and 2012" mentioned in the study's conclusion have actually occurred and are already known to us. So, what is the significance of the results shown by the model? In addition, the study predicts the mutation rates of fall and winter droughts in 2021 and 2022, which actually do not need to be predicted, but can be analyzed by directly consulting the test data. So what is the significance of the prediction here? I think we should predict the drought situation in the region after 2022 based on the previous data (at least it should be before 2022), such as the drought situation in the next five years, ten years, or even longer, so that it will have more realistic meaning for our agriculture. Otherwise, we get models or data that are only theoretical results.
Author Response
Thank you for the positive evaluation of our manuscript. Your constructive suggestions are very helpful for improving this manuscript Detailed responses to your comments are provided below. Hope you will find all your concern have been satisfactorily addressed.

Reviewer 2 Report
1. The author used a large space to introduce different drought indices and drought models. What is the weakness of these drought indices or models? Why you want to develop your own drought model, and what’s the novelty of your model? Did the author try to test the applicability of other indices or models in Kast regions?
2. In this paper, the author proposed the effects of evapotranspiration on agricultural drought many times. Since the author want to incorporate the influence of evapotranspiration, why did not use drought index developed using ET (e.g., SPEI)?
3. In this study, RF and SVR are used to construct drought models. The dependent variable is SSI. The author thought the drought index can represent the real agricultural drought, and based on the model results, the authors analyzed the spatiotemporal drought characteristics during a historical period (2001 - 2020). If the SSI can reflect agricultural drought, why did not use SSI that can directly be calculated using soil moisture data? What’s the purpose of this empirical modeling? I guess there are many soil moisture datasets released on the internet, both in-situ data and grided data, that can be used to calculate SSI.
4. The NDVI was used as an independent variable when constructing empirical models. Do you think NDVI/NDVIA is a result of SSI or the reason of SSI? The drought propagation follows this direction: precipitation -> soil moisture -> DNVI -> crop yield [1]. It’s ok to use NDVIA to predict soil moisture-based index, and you may get a good result. However, I want to know if the authors tried to consider the lag time between these input variables and how it can affect the model results since you considered different time scales here?
5. How did the author deal with the input data for different time scales? For example, for 3-month scale, did the author use accumulated precipitation for 3 months?
6. The author used model-derived time series (RF-CDI) to predict the drought conditions in the future several years. I am wondering if the author validated the predictions (for example, leave two years out when building prediction model and then use these two years to test the prediction results). I don’t think the predictions are reliable only based on time series data, especially when the variation of drought is insignificantly periodic. I do believe the SARIMA model may capture the period of El Nino events, but the increasingly frequent and uncertain extreme weather events significantly reduced the reliability of periodic forecasts, especially in recent several decades. Instead, did the author ever consider calculating drought index using projected climate data (e.g., HadGEM2-ES (https://esg.pik-potsdam.de/projects/isimip2a))?
Other suggestions:
1. “Domestic and foreign scholars” (Line52): change it to “many scholars”
2. Wrong typo: 1.76×105 km2, also seen in many other places (subscripts and superscript). Double check!
3. Using standard English names of months in Table 2&3
4. Table 3 – still is RF?
5. Missing figure and wrong citation of figures in paper
6. Figure 3 – give a more detailed caption. Is the correlation coefficient Pearson, Kendall, or Spearman?
7. Figure 4 – I suggest you use -0.5, -1, -1.5, and -2.0 in Table 1 to build contour lines.
8. Figure 8 – figure is unreadable. Please improve figure resolution.
9. Since you used many methods in your study, it is better if you can provide a flowchart.
10. Check the author’s name of papers you cited. Line 75: Ruping or Runping?
11. Double check your language.
Author Response
Thank you for the positive evaluation of our manuscript. Your constructive suggestions are very helpful for improving this manuscript. Detailed responses to your comments are provided below. Hope you will find all your concern have been satisfactorily addressed.

Reviewer 3 Report
April 09, 2023
Manuscript: 'Construction of an Agricultural Drought Monitoring Model for Karst with Coupled Climate and Substratum Factors -A Case 3 Study of Guizhou Province, China’
The manuscript explores the development of drought monitoring models for karst agriculture in southwest China, utilizing Random Forest and Support Vector Regression algorithms. The models incorporate multiple independent variables, including precipitation, evapotranspiration, NDVI, elevation, and karst development intensity, while the Standardized Soil Moisture Index serves as the dependent variable. The manuscript is well-structured and scientifically sound, falling within the scope of the MDPI water journal. Though a few minor revisions are necessary, I recommend publication of this article.
********************************
Title: this title accurately represents the content of the article.
Abstract: the abstract offers a comprehensive overview of the manuscript's content in a concise manner, and the selected keywords accurately represent the topics covered in the paper.
Lines 14-15: clarify the research gap and highlight the novelty of the study.
Line 26: write the meaning for the acronyms 'SARIMA' and 'RBEAST’; i.e., Seasonal Difference Autoregressive Moving Average for SARIMA.
Line 36: conclude with a brief statement on the practical implications of the study.
Introduction: the background information on agricultural drought monitoring is adequate, with relevant references and a discussion of their strengths and limitations, emphasizing the necessity of their assessment in Guizhou Province. The objectives and novelty of the study are clearly stated.
Lines 83-85: briefly explain the reasoning behind selecting precipitation, evapotranspiration, NDVI, elevation, and karst development intensity as independent variables in the study.
Lines 88-90: provide a brief explanation for the rationale behind choosing the Sen trend analysis and MK mutation tests to examine the attributes of agricultural drought.
Lines 90-91: the SARIMA and RBEAST models used in the study should be described in more detail, along with a clear justification for their selection.
Overview of the study area:
Lines 106-109: support this information with at least one quotation.
Materials and methods
Lines 121-124: indicate the thresholds used to divide the karst development intensity map by zones.
Lines 124-125: provide a concise rationale for the selection of precipitation, potential evapotranspiration, and NDVI as variables in the study.
Lines 166-167: Provide a brief explanation of the rationale behind selecting the RF and SVR regression algorithms for this task.
Results and analysis:
Line 416: to enhance clarity, replace 'Rbeast' with 'RBEAST'.
Discussion: this section is well-defined and substantiated by previous findings.
Conclusions: it is clear, concise, and in line with results and discussion.

Author Response
感谢您的支持评价。您的意见和建议对于改进我们的手稿至关重要。文章已根据您的建议仔细修改,如语法、插图、参考文献等。

Reviewer 4 Report
The manuscript introduces a model monitoring agricultural droughts coupling the geological conditions and climate. The karst area of Guizhou Province in south-West China was taken as the case study. Based on monthly data from January to December 2001-2020 the Authors attempted to predict the karst agricultural drought from January to December 2021–2025. In my opinion, the paper deserves attention as it may contribute to better prediction of droughts in areas built of carbonated rocks with widespread karst processes. I recommend the manuscript to be published in the journal. However, some amendments are required prior to the final acceptance of the paper for publication. Major flaws are as follows:
1. The text editing needs to be improved, including:
- p. 1, l. 18: “(…) Vegetation Index(NDVI),elevation, (…) – add spaces between the words;
- p. 1, l. 23: “(…), R2, (…)” – add “coefficient of determination (R2)”;
- in “R2” please change “2” into a superscript, as for example in l. 203, 206, 290, 292;
- please verify the notation of referencing numbers in the text (without superscripting), as for example on p. 2, l. 43: “damage[1];
- “2. Overview of the study area” – please change into “2. Study area”.
2. In the description of the study area please provide its more detailed description.
3. Figure 1b: DEM – colors referring to the elevation should be used in accordance with the cartographic standards, with brown representing mountains and green representing plains. Elevation (DEM) should be expressed in meters above sea level (m a.s.l.). Please correct.
4. P. 3, l. 116: “kg/m3” – “3” should be superscripted. Please correct.
5. Table 1: “light” correct into “Light”. The word “Drought” should be removed in each line of the second column.
6. Tables 2 and 3: the months numbering (1-12) need to be replaced with “Jan-Dec”.
7. Figure 3: information about p in pRF-CDI-1 is not shown in the figure caption, but it appears only in text (l. 306). Please correct. Moreover, please change “R-square” into “R2” in the horizontal axis labelling.
8. P. 12, l. 366: “95.506%” please change the notation to only one decimal “95.5%”; the same with other numbers in the paper.
9. Figure 6: the figure notation is too small and hardly legible. What is the reason of describing the respective sub-figures with (a) – (e)? It should be explained in the figure caption or it should be deleted. In the figure legend it would be more useful to describe the Mann-Kendall test significance by adding the p significance level, instead of describing it with “very sig.-rise” etc.
10. Figure 7. The respective subfigures and the explanations are too small and thus hardly readable. Please enlarge.
11. P. 14, l. 434: what is the meaning of “(86.475.675%)”? Please correct.
12. Figure 9 is of poor quality and the description of the vertical axis hardly visible. What is the meaning of the colors? Please correct.
Generally, it is recommended to accept the paper for publication after major revision.
Author Response
Thank you for your supportive evaluation. Your comments and suggestions were critical for improving our manuscript. The article has been carefully revised according to your suggestions, such as grammar, illustrations, references, etc.

Round 2
Reviewer 1 Report
I believe that the authorsm have made more comprehensive revisions in accordance with the comments I made, and that the paper in its current form meets water's requirements. Based on these, I recommend publication of the paper in the journal WATER.
Reviewer 4 Report
In my opinion explanations and corrections made by the Authors are satisfactory. I recommend to accept the paper for publication in present form.